# In Silico and In Vivo Pharmacological Evaluation of Iridoid Compounds: Geniposide and Asperuloside Profile Study Through Molecular Docking Assay and in the *Caenorhabditis elegans* Model

**DOI:** 10.3390/biom15081105

**Published:** 2025-07-31

**Authors:** Mariana Uczay, Péterson Alves Santos, Pricila Pflüger, Gilsane von Poser, José Brea, Maria Isabel Loza, Patrícia Pereira, José Angel Fontenla

**Affiliations:** 1GI-1684 Laboratory of Central Nervous System Pharmacology, Department of Pharmacology, Pharmacy and Pharmaceutical Technology, Faculty of Pharmacy, University of Santiago de Compostela, 15782 Santiago de Compostela, Spain; mariana.uczay@agronomicabr.com.br (M.U.); peterson.alves@ufrgs.br (P.A.S.); pricila.fernandes.pfluger@usc.es (P.P.); 2Laboratory of Neuropharmacology and Preclinical Toxicology, Institute of Basic Health Sciences, Federal University of Rio Grande do Sul, Porto Alegre 90050-170, Brazil; patriciapereira@ufrgs.br; 3Center for Research in Molecular Medicine and Chronic Diseases (CIMUS), University of Santiago de Compostela, 15782 Santiago de Compostela, Spain; pepo.brea@usc.es (J.B.); mabel.loza@usc.es (M.I.L.); 4Laboratory of Pharmacognosy, College of Pharmacy, Federal University of Rio Grande do Sul (UFRGS), Porto Alegre 91501-970, Brazil; gilsane.von@ufrgs.br

**Keywords:** neurodegeneration, iridoids, geniposide, asperuloside, molecular docking, *Caenorhabditis elegans*

## Abstract

Iridoids are compounds recognized for their neuroprotective properties and their potential application in the treatment of neurodegenerative diseases. Geniposide (GP) and asperuloside (ASP) are iridoids that have demonstrated some biological activities. In this study, the potential neuroprotective effects of these iridoids were evaluated through in silico and in vivo assays, using *Caenorhabditis elegans* (*C. elegans*) strains CF1553 (sod-3::GFP), GA800 (cat::GFP), and CL2166 (gst-4::GFP). The results suggested that neither compound appears to have good passive permeability through the blood–brain barrier (BBB). However, an active transport mechanism involving the glucose transporter GLUT-1 may be present, as both compounds contain glucose in their molecular structure. In addition, they can inhibit the activity of both acetylcholinesterase (AChE) and butyrylcholinesterase (BChE). GP at 1 and 2 mM reversed the H_2_O_2_-induced increase in sod-3 expression, while ASP at 1 and 2 mM reversed the increase in gst-4 expression. Worm survival was more adversely affected by higher concentrations of GP than ASP, although both similarly reduced acetylcholinesterase activity. These findings suggest that GP and ASP exhibit very low toxicity both in silico and in vivo in *C. elegans*, and positively modulate key enzymes involved in antioxidant pathways, highlighting their potential for neuroprotective applications.

## 1. Introduction

Diseases of the central nervous system (CNS) remain complex, despite the advances made in recent years in understanding their pathophysiological mechanisms. Brain pathologies, such as Alzheimer’s disease (AD), Parkinson’s disease (PD), and stroke, are becoming increasingly common as the life expectancy of the world’s population increases, and cause marked disability, necessitating the development of new treatments [1].

Alzheimer’s (AD) and Parkinson’s (PD) are the two most important neurodegenerative diseases, having a substantial individual and social impact [2]. Despite the growing population of patients with AD, only the pharmacological options listed below are currently used to treat the symptoms of AD: memantine, a N-Methyl-D-aspartate (NMDA) glutamate receptor antagonist; the cholinesterase inhibitors (ChEIs) donepezil, galantamine, and rivastigmine; and the combination of memantine–donepezil [3]. Although there are treatments to alleviate AD’s mild symptoms, due to its polygenic nature, no drug is currently in use that significantly improves cognition or prevents the progression of AD [4].

In 2021, the U.S. Food and Drug Administration (FDA) approved the first monoclonal antibody, aducanumab (Aduhelm^®^), for AD. This drug was withdraw by the laboratory Biogen in 2024 after several scientific requests due to its ineffectiveness and potential harm. Lecanemab is the second monoclonal antibody, which was approved in 2022 by the FDA under the accelerated procedure and in 2024 by the European Medicines Agency (EMA) except for ApoE4 homozygotes. Studies of other monoclonal antibodies against amyloid peptide (AB) or hyperphosphorylated Tau protein are in progress.

AD is among the least well-served therapeutic areas of drug treatments and the needs of patients are not adequately addressed, so further research is needed. Emerging evidence suggests adenosine G protein-coupled receptors are promising therapeutic targets for AD. The Adenosine A_1_ and A_2A_ receptors are expressed in different areas of the human brain and have a proposed involvement in the pathogenesis of dementia [5] and PD. In certain countries such as, for example, Japan, the United States of America, and Canada, istradefylline (an A_2A_ receptor antagonist) has been approved for the treatment of PD, but the EMA has denied its approval.

PD is the second most important neurodegenerative disease, characterized by motor and nonmotor symptoms [6]. Symptoms are related to the loss of dopaminergic neurons from the *pars compacta* of the substantia nigra (*pc*SN) [7], and include motor and cognitive deficits such as tremor, rigidity, bradykinesia, psychiatric abnormalities, smell and speech disturbance, and other pathological implications [8]. Dopaminergic facilitation (L-dopa or L-dopa plus dopa decarboxylase inhibitor, monoamine oxidase-B (MAO-B) and Catechol-O-methyltransferase (COMT) inhibitors, and dopamine receptor agonist) is still used to provide symptomatic relief, but, in most cases, these drugs do not promote neuroprotection [9]. The anticholinergic approach in the treatment of PD is the most classic and presents several adverse effects (blurred vision, urinary retention, constipation, etc.). Drugs such as benztropine, trihexyphenidyl, biperiden, and other anticholinergics continue to be used clinically for the treatment of PD [10].

Geniposide (GP) is one of the main iridoid glycosides; it has been isolated from almost 40 plant species, and most of them are used as traditional herbal medicines in Chinese medicine. The pharmacological activities of GP include neuroprotective, analgesic, antidepressant, anti-inflammatory, antioxidant, immunoregulatory, and antithrombotic functions, as well as hepatoprotective and antitumoral effects [11,12]. The known mechanisms of GP are related to the increased activity of antioxidant enzymes such as superoxide dismutase (SOD) and catalase (CAT), as well as the inhibition of reactive oxygen species (ROS) and nitric oxide (NO), in addition to its regulatory effects on signaling pathways such as AMP-activated protein kinase (AMPK) and phosphatidylinositol 3-kinase/Akt (PI3K/Akt) [13,14,15].

Many studies related to GP have focused on its therapeutic potential in the CNS. An in vitro study with GP showed cytoprotective effects in rat hippocampal cell lines, reducing neuronal cell death induced by oxygen and glucose deprivation [16]. Additionally, GP acts on the signaling of Bax/Bcl-2 proteins, thereby reducing the process of neuronal apoptosis [17]. In an experimental Alzheimer’s model, GP suppressed the overactivation of MAPK signaling, leading to a reduction in the accumulation of amyloid-β (Aβ) and an improvement in cholinergic deficits in the hippocampus [18]. In another study, GP exerted protective effects against Aβ-induced injury, attenuating neuronal damage caused by oxidative stress [19]. Moreover, in other pathologies such as epilepsy, which is increasingly associated with AD and PD, GP attenuates electroshock-induced epileptic seizures, significantly reducing their incidence and increasing the latency of clonic seizures [15].

Another iridoid with a chemical structure like GP, which also has notable biological effects, is asperuloside (ASP). This compound is known to exhibit pharmacological properties such as anti-obesity activity [20]. In addition, ASP has been shown to significantly reduce levels of nitric oxide and prostaglandins in an inflammation model, exerting an anti-inflammatory effect [21]. However, few studies have been conducted on the activity of this compound, and so far, no experimental studies have been carried out associating ASP with its potential neuroprotective effect. The lack of literature on ASP, particularly its neuroprotective potential, led to its inclusion in this study to explore its therapeutic relevance for central nervous system disorders.

Therefore, considering the various pharmacological activities already reported for ASP and GP, this study aimed to investigate the potential mechanisms of action by which these compounds may be useful in the treatment of central nervous system (CNS) disorders. Additionally, we aimed to evaluate their toxicity profiles. To achieve these goals, we conducted both in silico and in vivo assays, including molecular docking analyses and experiments using the *Caenorhabditis elegans* model.

## 2. Materials and Methods

### 2.1. In Silico and Docking Studies

#### 2.1.1. Prediction of Physicochemical and ADME Properties

To understand the properties of GP, ASP, and the reference drugs used for the treatment of AD and PD, the 2D structures of the compounds were obtained using ChemDraw software (Ultra version 21.0, PerkinElmer Informatics, Waltham, MA, USA) and converted into a single SMILES database file. In silico predictions were made using the Molinspiration platform (Molinspiration Cheminformatics software, https://www.molinspiration.com/, Slovenský Grob, Slovak Republic) and the pkCSM tool (https://biosig.lab.uq.edu.au/pkcsm/prediction (accessed on 28 July 2025), Brisbane, Australia) to evaluate the compounds’ absorption, distribution, metabolism, and excretion (ADME) properties and to compare these results with those of reference drugs used for AD and PD.

#### 2.1.2. Prediction of Ability to Cross the BBB

The ability of compounds to cross the BBB was assessed using the Online BBB Predictor (https://www.cbligand.org/BBB/, (accessed on 28 July 2025) Pittsburgh, PA, USA), employing two algrithms—support vector machine (SVM) and AdaBoost—alongside four types of molecular fingerprints: MACCS, Open Babel (FP2), Molprint 2D, and PubChem [22].

#### 2.1.3. Prediction of the GP and ASP Transport Mechanism at BBB

To carry out predictions of GP and ASP transport mechanisms, we used molecular docking assays. As these compounds are glycoside iridoids and contain glucose molecules in their structure, we performed docking analyses using the CB-Dock2 platform (see Section 2.1.4) to assess their binding affinity for the main mechanisms of glucose transport across BBB into the CNS. The D-glucose molecule was obtained from the PubChem database, and the GLUT-1 receptor structure from the Protein Data Bank (PDB ID: 5EQI; resolution 3.0 Å).

#### 2.1.4. Docking Studies

Docking analyses were conducted using the HTDocking platform (https://www.cbligand.org/AD/docking_search.php (accessed on 28 July 2025)), an online tool that automates the docking process to identify potential protein targets. In this system, the docking score is expressed as pKi, where pKi = –log (predicted Ki) and predicted Ki = exp (ΔG × 1000/(1.9871917 × 298.15). Each compound’s docking score against a given protein structure is used to evaluate and rank its possible protein partners. A strong docking score (above 6.0 or predicted Ki < 1000 nM) suggests that the protein may be a likely target for the tested small molecule.

Next, we analyzed the proteins showing the highest binding affinities as predicted by the CB-Dock2 platform (https://cadd.labshare.cn/cb-dock2/php/index.php (accessed on 28 July 2025), Chengdu, China), a blind docking tool that integrates curvature-based cavity detection with AutoDock Vina-based molecular docking (V 1.1.2). The crystal structures of the proteins of interest were obtained from the Protein Data Bank (https://www.rcsb.org/). Specifically, we employed acetylcholinesterase (AChE, EC 3.1.1.7) (PDB ID: 4EY6; resolution: 2.40 Å) and butyrylcholinesterase (BChE, EC 3.1.1.8) (PDB ID: 4TPK; resolution: 2.70 Å). GP, ASP, and galantamine—the reference ligand used as an anticholinesterase control—were designed using the Marvin JS platform (https://marvinjs-demo.chemaxon.com/latest/demo.html (accessed on 28 July 2025), V 25.1.0, Budapest, Hungary), and converted into 3D structures in Avogadro software (https://avogadro.cc/, (accessed on 28 July 2025) Pittsburgh, PA, USA) using the Open Babel 2.3.2 tool (https://www.cheminfo.org/Chemistry/Cheminformatics/FormatConverter/index.html (accessed on 28 July 2025)).

### 2.2. In Vivo Studies in Caenorhabditis elegans

#### 2.2.1. Compounds

GP was extracted, by chromatographic methods, from *Genipa americana* and ASP from *Escallonia bifida* and *Escallonia megapotamica*. Both compounds were isolated from plant sources using standard column chromatography techniques. Extracts were subjected to repeated open-column chromatography, using silica gel to fractionate and purify individual iridoids. The compounds were extracted in the Pharmacognosy laboratory in the Faculty of Pharmacy of the Federal University of Rio Grande do Sul (UFRGS), Brazil. The structure of the compounds was confirmed by high-resolution mass spectrometry [23].

#### 2.2.2. Strains and Maintenance of C. elegans

Wild-type Bristol (N2), CF1553 (sod-3::GFP), GA800 (cat::GFP), and CL2166 (gst-4::GFP) strains of *C. elegans* were grown on plates with Nematode Growth Media (NGM) seeded with a lawn of *Escherichia coli* strain OP50 (*E. coli*-OP50) as a food source and maintained under conditions described previously. Regular subculturing was performed to prevent overcrowding and to maintain the *C. elegans* population. For all experiments, adult hermaphrodites were synchronized using an alkaline hypochlorite solution (1 M NaOH, 5% NaClO). The L4 larval stage was used and maintained at 20 °C in the dark. In all experimental analyses, investigators were blinded to the treatment groups, and coded identifiers were used during statistical analysis to avoid experimental bias. Animals used in the assays were randomly assigned to treatment conditions.

#### 2.2.3. Antioxidant Properties

Strains CF1553 (sod-3::GFP), GA800 (cat::GFP), and CL2166 (gst-4::GFP) were synchronized with hypochlorite solution and incubated at 20 °C for 48 h. Then, the L4 stage larvae were exposed to GP or ASP (0.5, 1 or 2 mM), hydrogen peroxide (2 mM), or K-medium (control) for one hour in 24-well plates containing liquid medium. Then, the worms were anesthetized with 0.5 mM sodium azide and transferred to histological slides and covered with coverslips. The slides were analyzed under a fluorescence microscope Olympus BX41 at 10× magnification (Olympus Iberia S.A.U., Spain). Ten worms per treatment (N= 10) were analyzed, and individual experiments for each group were performed in triplicate. Images were analyzed using NIH ImageJ software (V 1.54j).

#### 2.2.4. Body Length Assay

The body length assay for GP and ASP was performed in 24-well plates with a final volume of 0.3 mL of liquid medium in the presence of food (2 μL of *E. coli*-OP50 suspension). Ten age-synchronized L1 worms (N = 10) were transferred to each well, which contained serial dilutions of GP or ASP (0.5, 1, and 2 mM) or K-medium only (control).

The worms were maintained under the same conditions until the L4 larval stage. After that, images of each worm were captured using a Motic BA310 microscope (Motic Europe, Barcelona, Spain) at 10× magnification. The body size was then analyzed using Motic Image Plus 3.0 software. Experiments were conducted in triplicate.

#### 2.2.5. Lethality Test and LC_50_ Estimation

The lethality assays for GP, ASP, and L-dopa were conducted in 24-well plates with a final volume of 1 mL of liquid medium containing food (20 μL of *E. coli*-OP50 suspension). Ten age-synchronized L4 worms (N= 10) were transferred into each well, which contained serial dilutions of GP or ASP (0.1, 0.5, 1, 2, and 5 mM) or L-dopa (2 and 5 mM). The concentration ranges used in the toxicity tests were selected based on previous in vitro studies [23,24,25,26]. The negative controls (just the K-medium) were included in triplicate on each plate. Plates were incubated in the dark, for 24 h, at 20 °C. After exposure, live and dead worms were counted by visualization under a microscope. Nematodes were considered dead if they did not respond to stimulus when touched with a metal wire. Results were expressed as the percentage of live worms of the total number of nematodes in each well (survival rate). Data were presented as the mean ± SEM of three independent experiments.

#### 2.2.6. Anticholinesterase Activity Assay

Approximately 1000 N2 worms were incubated with GP or ASP (1, 2, 5, or 10 mM). After 24 h, the worms were collected using M9 buffer and sonicated. The lysate was cleared by centrifugation for 15 min at 12,000 rpm and 4 °C.

Cholinesterase activities (AChE and BChE) were measured according to the instructions of the respective commercial kits (Bioclin), and absorbance was read at 405 nm with a high-performance multimode detection reader [27].

#### 2.2.7. Graphical Representation and Statistical Analyses

The graphical and statistical analyses were performed using GraphPad Prism software (V 8.0.1) (GraphPad Software, Boston, MA, USA). The minimum sample size for each endpoint was determined to ensure a normal distribution of the data. The lethal concentration (LC_50_) value for each compound was calculated using non-linear regression.

The differences observed between nematodes exposed to the test solutions and the control were analyzed using one-way ANOVA, followed by Dunnett’s test. Results are expressed as the mean ± standard error of the mean (SEM) from at least three replicates. Differences were considered statistically significant at *p* ≤ 0.05.

## 3. Results

### 3.1. In Silico and Docking Studies

The molecular structures of geniposide (GP) and asperuloside (ASP), along with their corresponding SMILES representations used in pharmacological and molecular docking analyses, are shown in Figure 1.

#### 3.1.1. Prediction of Physicochemical and ADME Properties

The predicted parameter values, including lipophilicity (expressed as miLogP), molecular weight (MW), number of hydrogen bond acceptors (nON), number of hydrogen bond donors (nOHNH), and the number of violations of Lipinski’s Rule of Five (NV), are presented in Table 1.

GP and ASP show low miLogP values (–1.53 and –2.65, respectively), like L-dopa (–2.20). However, it should be noted that L-dopa is absorbed via an active transport mechanism (neutral amino acid transporter) [28]. All other reference drugs evaluated present positive miLogP values, ranging from 1.54 to 4.10, for galantamine and donepezil, respectively.

The molecular weight of GP (388.37 g/mol) and ASP (414.36 g/mol), like the other reference drugs, is below 500 g/mol, although they are among the highest values of all the drugs evaluated. Lipinski’s rules are not violated by geniposide, but asperuloside has eleven hydrogen bond acceptors (nON > 10), causing it to violate one of the rules.

Molecular volume (MV) determines characteristics of molecules, such as intestinal absorption or BBB penetration. Calculated volume is expressed in cubic Angstroms (A^3^). GP has a MV of 331.54 and ASP of 340.68, being in the highest zone of the MV (Table 2).

GP, ASP, and the other evaluated drugs show negative water solubility values. These values are between −4.97 (donepezil) and −2.64 (galantamine). GP has a predicted water solubility of −2.81 and ASP of −3.27 (Table 2).

The topological polar surface area (TPSA) is the sum of the surfaces of polar atoms in a molecule (typically oxygen, nitrogen, and their attached hydrogen). L-dopa, GP, and ASP show values above 100 Å^2^.

Nrotb is a measure of the molecule’s flexibility. The lower the better (see Discussion section). GP, ASP, and donepezil showed the highest values.

#### 3.1.2. Prediction of Ability to Cross the BBB

For a drug to be effective in CNS diseases, it must cross the BBB and reach adequate concentrations at its site of action. Therefore, we assessed whether GP and ASP, along with the most used drugs for AD and PD, are predicted to have good CNS permeability. Negative values indicate poor predicted permeability through the BBB. The values obtained for GP and ASP do not support adequate passage through the BBB by passive diffusion, like L-dopa, which also does not access the CNS by passive diffusion (Table 3). Low BBB permeability results in insufficient drug concentrations in brain targets, which is considered one of the main reasons for the failure of CNS-targeted drug therapies [29,30].

#### 3.1.3. Prediction of the GP and ASP Transport Mechanism

We hypothesized that GP and ASP, as they contain glucose moieties in their structures (Figure 1), could cross the BBB via the same transport mechanisms used by glucose. To test this, we evaluated in silico the binding affinity of these compounds with the main glucose transporter, GLUT-1. The results can be seen in Table 4 and Figure 2. GP and ASP exhibit similar binding profiles and interact with the same amino acid residues as D-glucose.

GP, ASP, and L-dopa are predicted to be P-glycoprotein (P-gp) substrates by the in silico analysis. As with most of the evaluated drugs, only donepezil is predicted to be a P-gp inhibitor (Table 5). Drugs that are P-gp substrates but not inhibitors tend to have lower permeability to the CNS.

#### 3.1.4. Docking Studies

Docking analyzes using the HTDocking program (Table 6) revealed that GP has an affinity for acetylcholinesterase (AChE) and butyrylcholinesterase (BChE) enzymes, with Ki scores of 9.3 and 9.2, respectively. And ASP has a Ki value of 9.5 for AChE and 9.4 for BChE. For the reference drug, galantamine, the values were 9 and 8 for AChE and BChE, respectively; results like those were found in the literature [31].

Therefore, we decided to analyze the binding energy of these two compounds with these proteins thought the CB-Dock2 program. In the in silico analysis, GP binds on the same pocket as galantamine on the active site of the AChE with a binding energy of −8.8 kcal/mol (Table 7). GP binds with various AChE residues also in a similar manner to galantamine, such as tryptophan residue 86 and glycine 120, 121, and 122. Furthermore, both share a hydrogen bond at the serine residue 203.

ASP also binds with AChE on same site as galantamine with a binding energy of −8.0 kcal/mol. ASP interacts with AChE at aspartate residue 74, via hydrogen bonds, and with glycine residues 120, 121, and 122, as well as galantamine. Additionally, ASP binds to BChE with a free energy of −8.7 kcal/mol (Figure 3).

### 3.2. In Vivo Studies in C. elegans

#### 3.2.1. Antioxidant Properties of GP and ASP

Figure 4 presents the relative expression of GST, SOD, and CAT, after exposure to K-medium (control), hydrogen peroxide (H_2_O_2_, as positive control), or hydrogen peroxide plus GP (0.5, 1 or 2 mM). H_2_O_2_ increased GST expression, but GP 0.5 mM was able to partially reverse the damage, decreasing its expression. In the evaluation of SOD, hydrogen peroxide increased its expression, but GP (1 and 2 mM) reduced SOD activity. None of the GP concentrations altered CAT expression after exposure to the stressor.

All concentrations of ASP (0.5, 1, and 2 mM) were able to reduce the damage caused by hydrogen peroxide in the GST enzyme (Figure 5). As for SOD and CAT enzymes, ASP had no effect.

#### 3.2.2. Body Size Assay

Figure 6 shows the results of the body size assay of worms treated with GP and ASP. Only ASP 0.5 and 1 mM slightly decreased the worm’s body size. The body width also was evaluated, and no changes were observed in the worms treated with the compounds.

#### 3.2.3. Survival Assay and LC_50_

The survival results are shown in Figure 7. The compounds exhibited a dose-dependent effect on adult nematodes mortality after 24 h of exposure with LC_50_ 2.47 and 4.58 mM, respectively. In all cases, survival is higher with GP and ASP than with L-dopa.

#### 3.2.4. Anticholinesterase Activity

Figure 8 shows the measurement of anticholinesterase enzyme activity. Both higher concentrations of ASP and GP (5 and 10 mM) significantly reduced enzyme activity.

## 4. Discussion

The development of efficient therapies for neurodegenerative diseases remains a significant challenge. The use of natural products has been widely acknowledged as highly promising and is often considered a safer alternative to synthetic drugs. Currently, in silico studies provide essential preliminary data on potential drug candidates, which can greatly assist in designing subsequent in vitro and in vivo experiments.

To model the bioavailability of new pharmacological active substances for their potential use as orally administered drugs, a common approach is to screen compounds based on Lipinski’s Rule of Five. This rule states that a drug is more likely to exhibit good absorption or penetration if its chemical structure does not violate more than one of the following criteria: (1) miLogP (octanol-water partition coefficient) ≤ 5; (2) molecular weight (MW) ≤ 500 Da; (3) number of hydrogen bond acceptors (nON) ≤ 10; (4) number of hydrogen bond donors (nOHNH) ≤ 5.

The octanol-water partition coefficient logP is used as a measure of molecular hydrophobicity, which influences drug absorption, bioavailability, hydrophobic drug–receptor interactions, molecular metabolism, and toxicity.

miLogP is a logP prediction method developed by Molinspiration (miLogP2.2). GP and ASP exhibit negative miLogP values, like L-dopa. As previously mentioned, L-dopa relies on an active transport mechanism, the large neutral amino acid transporter (LAT1), for its intestinal absorption or for BBB crossing [28]. On the contrary, all other drugs evaluated show positive miLogP values.

The molecular weight of GP (388.37 g/mol) and ASP (414.36 g/mol), like the other reference drugs, is below 500, but they are in the highest range of all the drugs evaluated. Drugs which are active in the CNS have significantly reduced MW compared with drugs that act in other body zones. In general, molecules that are between 400–500 g/mol have good permeability to the CNS [32]. GP and ASP meet the requirement of having a molecular weight within the valid range for accessing the CNS.

Compounds with high hydrogen bond forming potential have minimal distribution through the BBB [33]. The marketed CNS drugs on average have 2.12 hydrogen bond acceptors (nON) and 1.5 hydrogen bond donors (nOHNH) [32]. GP and ASP have a very high number of both parameters (10 and 11 nON and 5 and 4 nOHNH, respectively).

GP meets the requirements of Lipinski’s Rule of Five to be considered a good drug. ASP has a higher-than-expected number of hydrogen bond acceptors, and therefore violates one of the rules. However, Lipinski’s Rule of Five is irrelevant to those compounds that may more readily cross the biological membranes through the actuation of the active transport systems. For instance, natural products and compounds derived from natural products are frequently recognized as substrates by biological transporters, thereby enabling them to circumvent the constraints imposed by Lipinski’s Rule of Five on intestinal wall permeability [34].

Based on single-conformation TPSA calculations, it appears that the conformational range is limited in drugs that act on the CNS. GP, ASP, and L-dopa fall into the highest TPSA range among all compounds evaluated, with values of 155.15, 161.22, and 103.68 Å^2^, respectively. This TPSA parameter has been shown to correlate strongly with human intestinal absorption, permeability in Caco2 monolayers, and BBB penetration. Drugs with a TPSA of 60 Å^2^ or less are fully absorbed, whereas those with TPSA ≥ 140 Å^2^ are typically not absorbed efficiently [35]. Both GP and ASP have TPSA values greater than 140 Å^2^.

Compounds with more than ten rotatable bonds (Nrotb) are correlated with reduced oral bioavailability in rats. CNS drugs tend to have significantly fewer rotatable bonds than other classes of drugs. GP, ASP, and donepezil each possess six rotatable bonds in their molecular structures. Most centrally acting compounds that cross the BBB by simple diffusion have a rotatable bond count of five or fewer [35].

The colorectal adenocarcinoma cells (Caco2) are used as a predictive model to assess gastrointestinal permeability. The greater the cellular permeability, the more likely a compound is to be absorbed in the gastrointestinal tract. The lowest predicted values were observed for L-dopa, GP, and ASP (Table 2).

Most drugs will exist in equilibrium between either an unbound state or bound to blood proteins. The diffusion potential of a drug is affected by the degree to which it binds to blood proteins, since the more it binds, the less free fraction is available to cross cell membranes. GP, ASP, and L-dopa have the highest values for percentage not bound to proteins.

The volume of distribution (VD) refers to the theoretical volume in which the total administered dose of a drug would need to be uniformly distributed to achieve the same concentration as that observed in blood plasma. The higher the VD (log L/kg), the more of a drug is distributed in tissue rather than in plasma. The compounds that have the lowest predicted VD values are GP (−0.22), ASP (−0.02), and L-dopa (−0.10), and the highest is donepezil (1.26) (Table 2).

The high percentage of GP and ASP that does not bind to blood proteins and lower values of the distribution volume (VD) indicate that GP and ASP cannot diffuse freely (simple diffusion) through the membranes. This is also in agreement with the low value of miLogP calculated, which is a clear indication of the greater predilection of GP and ASP for the aqueous medium of blood *versus* the lipid medium of membranes.

Drugs clearance is measured by the proportionality constant called total clearance (C tot, log ml/min/kg), and occurs as a combination of hepatic clearance and renal clearance. The total clearance values of GP and ASP were the highest of all molecules, indicating the ease with which they are eliminated from the organism (Table 2).

In the evaluation of the blood–brain-barrier-crossing capacity, positive values indicate that the compounds can cross by passive diffusion through the BBB, whereas negative values indicate poor permeability. GP and ASP have negative values in five of the eight calculated parameters, while L-dopa has negative values in seven of the eight calculated parameters. On the other hand, the rest of the molecules always presented positive values in all parameters. Despite the theorical predicted values for the passage of GP and ASP through the BBB, and consequently their access to the CNS, in previous studies, CNS effects have been observed in mice or rats treated with these compounds in models of CNS or inflammatory diseases [15,36,37]. Furthermore, a micro dialysis study detected GP in the hippocampus of rats after intragastric administration of Zhi-zi-chi decoction (a solution used for the treatment of depression for centuries in China) [18].

A review study analyzed the antidepressant activity of the main iridoids found in the literature, noting that GP has several beneficial effects in models of depression in rats and mice [38]. Since these aforementioned in vivo studies have shown that GP and ASP have access to the CNS, and we have found that they theoretically do not readily access the brain by passive diffusion, we hypothesized that, as well as L-dopa, there may be a transport system in the BBB that delivers GP and ASP to the brain.

The development of new strategies to overcome the BBB is considered essential for more effective therapies. GP and ASP, being iridoid glycosides, have glucose residues in their molecules. Several studies use the strategy of conjugating drugs with glucose molecules to improve their passage through the BBB [15,39], with promising results. In the study by Temizyürek et al. [40], they observed that the conjugate of lacosamide with glucose molecules improved the access of the drug to the brain in a rodent model with epileptic seizures. In silico, GP and ASP have a similar and high binding affinity for the GLUT-1 transporter, interacting with the same amino acid residues as D-glucose; therefore, this may indicate how these compounds can cross the BBB using this transport system, but further studies are still needed to know whether this transporter can take the compounds to the CNS.

P-glycoprotein (P-gp) has been identified in the brain, exhibiting high levels of expression on the luminal (blood-facing) side of the brain vascular endothelium, as well as in the endothelial cells of the choroid plexus, where it facilitates transport toward the cerebrospinal fluid. Notably, CNS drugs have been observed to exhibit a significantly lower incidence of P-gp-mediated efflux compared to non-CNS drugs, indicating the greater relevance of P-gp efflux for CNS therapeutics. This is likely because, in contrast to the intestine, drug plasma concentrations at the BBB rarely reach levels sufficient to saturate P-gp [41]. A study evaluated the potential of GP as an inhibitor of multiple drug resistance by the overexpression of P-gp, evaluated in MG66 cells; there was a greater action of doxorubicin, increasing its action. This may be due to competitive inhibition, or a decrease in P-gp expression [42]. P-gp is an important protein involved in the efflux of drugs out of cells, mainly in the BBB, and intestinal epithelial cells. GP, ASP, and L-dopa are predicted to be P-gp substrates, as were most of the evaluated drugs, but only donepezil is predicted to be P-gp inhibitor (Table 5).

ASP also binds with AChE on same site as galantamine with a binding energy of −8.0 kcal/mol (Table 7). ASP interacts with AChE at aspartate residue 74, via hydrogen bonds, and with glycine residues 120, 121, and 122, as well as galantamine. Additionally, ASP binds to BChE with a free energy of −8.7 kcal/mol (Figure 3). Both ASP and GP reduced cholinesterase activity in N2 worms (Figure 8), leading to the suggestion that both compounds may have cholinergic neurotransmission-enhancing activity, a fact that is corroborated by the high binding affinity of these compounds to the enzymes AChE and BChE (Table 7 and Figure 3). The in vivo effect of GP on AChE has already been evaluated in an Alzheimer’s model in mice APP/PS1, and it was observed that GP (50 mg/kg) was able to attenuate the cholinergic deficit in these animals, increasing choline acetyl transferase (ChAT) and decreasing AChE [18]. Also, GP binds with a binding affinity of −9.1 kcal/mol in BChE in the same pocket of galantamine, in the residues of aspartate 70, tyrosine 114, and glycine 116 and 439.

GP was effective at minimizing the harm caused by hydrogen peroxide in a *C. elegans* strain that had been modified to express the fluorescent enzyme SOD, demonstrating an antioxidant activity. The antioxidant activity of GP has already been evaluated in a study that evaluated rotenone-induced damage in mice; GP at doses of 25 and 50 mg/kg significantly reduced the GSH:GSSG ratio [43]. In another study, GP could decrease oxidative-stress-induced apoptosis in HK-2 cells and normalize the activity of SOD1 and SOD2 enzymes [44].

ASP demonstrated antioxidant activity by reducing levels of gst-4 in a *C. elegans* strain that was designed to express the fluorescent enzyme. Previously, ASP has been evaluated for its antioxidant activity in RAW 264.7 cells, demonstrating a reduction in oxidative stress after induction of damage by lipopolysaccharide [45].

In *C. elegans*, phenotypic changes are typically associated with toxicity induced by various poisons or stresses [46]. Body length serves as a classic endpoint for the assessment of nematode toxicity [47]. Worms were treated with GP and ASP, and only ASP-treated worms showed a slight decrease in body size at two lowest concentrations, compared to the untreated control. We also evaluated the body width of the treated worms, which did not change in any of the evaluated concentrations.

The survival of the worms in 24 h was not affected by both compounds tested at lower millimolar concentrations (0.1 and 0.5 mM). However, higher GP concentrations (0.5 mM and above) increased mortality. The test showed that, in the tested concentration, ASP was less toxic because only the highest concentration could affect the survival of *C. elegans* in 24 h (Figure 7).

Although the findings presented here suggest potential effects related to blood–brain barrier permeability and neuroprotective activity, it is important to interpret these results within the context of the experimental model used. *C. elegans* is a well-established system for preliminary in vivo screening, offering insights into neurobehavioral and toxicological responses; however, its anatomical and physiological differences from mammalian systems limit the direct extrapolation of certain outcomes, particularly those involving complex structures such as the BBB. Nonetheless, the observed responses provide initial evidence that supports the biological relevance of the compound and lays the groundwork for future studies using vertebrate models. These additional investigations will be essential to confirm the mechanisms proposed and to more precisely assess the compound’s pharmacological potential in higher organisms.

## 5. Conclusions

Geniposide and asperuloside can cross the BBB, possibly through the glucose transporter GLUT-1. Furthermore, GP and ASP have a neuroprotective profile and acetylcholinesterase inhibition capacity. In silico and in vivo studies have shown that these two iridoids are safe molecules and have a lower toxicity profile than L-dopa. Taking together, the results obtained corroborate the initial idea that ASP and GP are potential candidates for the treatment of neurodegenerative diseases.

## Figures and Tables

**Figure 1 biomolecules-15-01105-f001:**
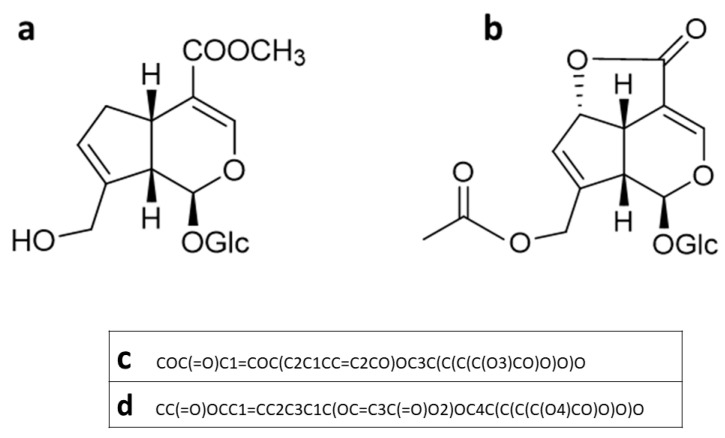
Molecular structure of geniposide (**a**) and asperuloside (**b**) drawn in ChemDraw (version Ultra 21.0, PerkinElmer Informatics, Waltham, MA, USA) and SMILES code used for pharmacological and molecular docking analyses of geniposide (**c**) and asperuloside (**d**).

**Figure 2 biomolecules-15-01105-f002:**
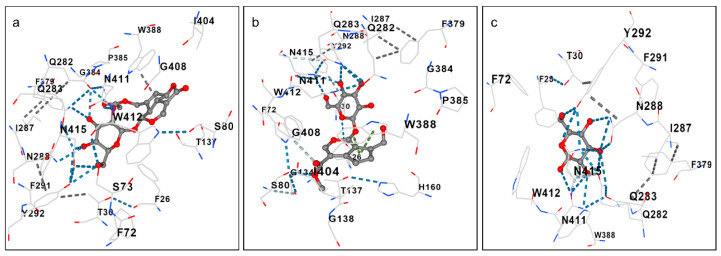
Predicted interactions between GP (**a**), ASP (**b**), and D-glucose (**c**) with GLUT-1.

**Figure 3 biomolecules-15-01105-f003:**
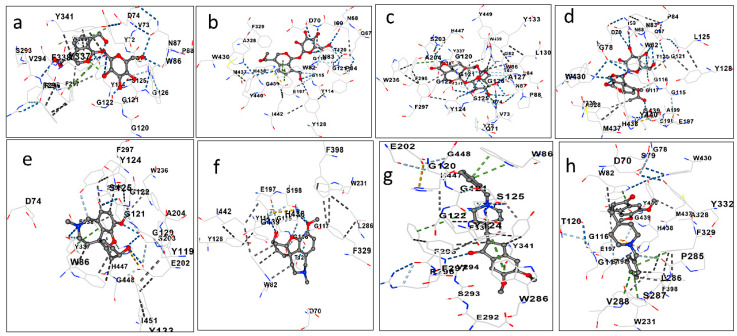
Predictions of GP interactions with AChE (**a**) and BChE (**b**), ASP with AChE (**c**) and BChE (**d**), galantamine with AChE (**e**) and BChE (**f**), and donepezil with AChE (**g**) and BChE (**h**).

**Figure 4 biomolecules-15-01105-f004:**
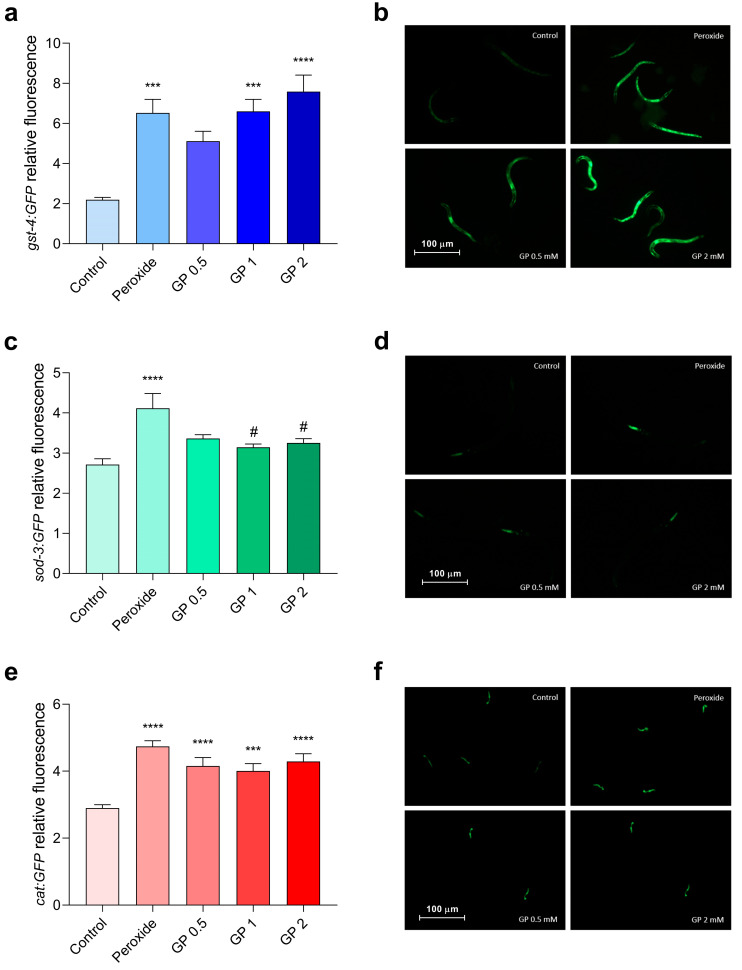
GST (**a**,**b**), SOD (**c**,**d**), and CAT (**e**,**f**) relative expression in *C. elegans* after K-medium (control), hydrogen peroxide only exposure (positive control), or GP + hydrogen peroxide. N = 10 worms. *** *p* < 0.001, and **** *p* < 0.0001 versus control group. # *p* < 0.05 versus hydrogen peroxide group. Values represent means ± SEM. Analyzed by one-way ANOVA followed by Dunnett’s test.

**Figure 5 biomolecules-15-01105-f005:**
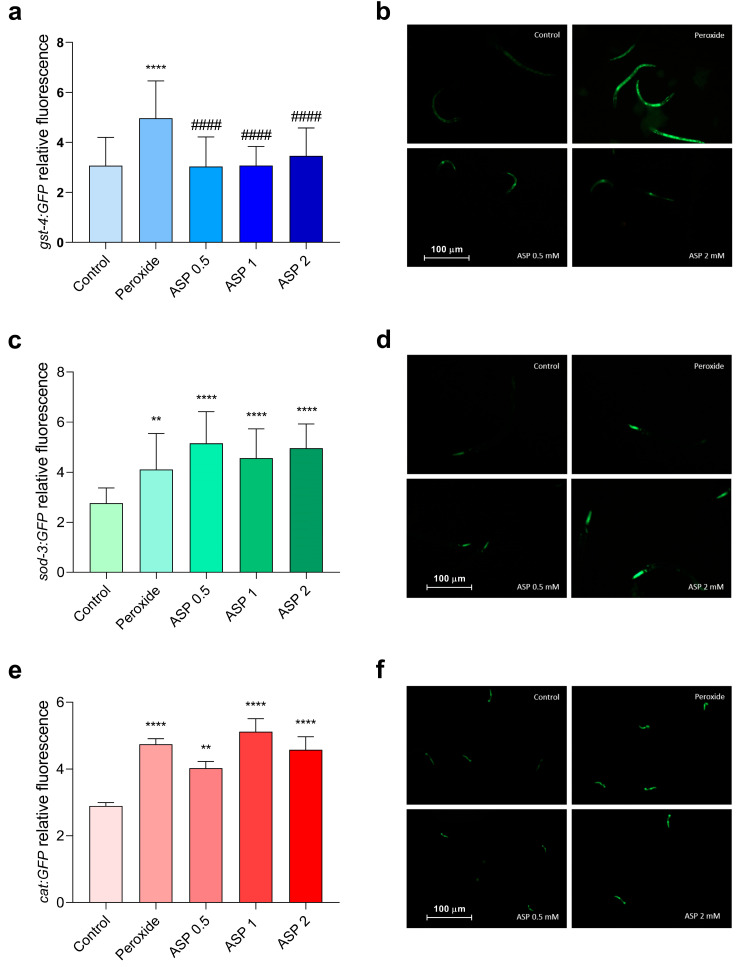
GST-4 (**a**,**b**), SOD-3 (**c**,**d**), and CAT (**e**,**f**) relative expression in *C. elegans* after K-medium (control), hydrogen peroxide (positive control), or ASP + hydrogen peroxide. N = 10 worms. ** *p* < 0.01, and **** *p* < 0.0001 versus control group. #### *p* < 0.0001 versus hydrogen peroxide group. Values represent means ± SEM. Analyzed by one-way ANOVA followed by Dunnett’s test.

**Figure 6 biomolecules-15-01105-f006:**
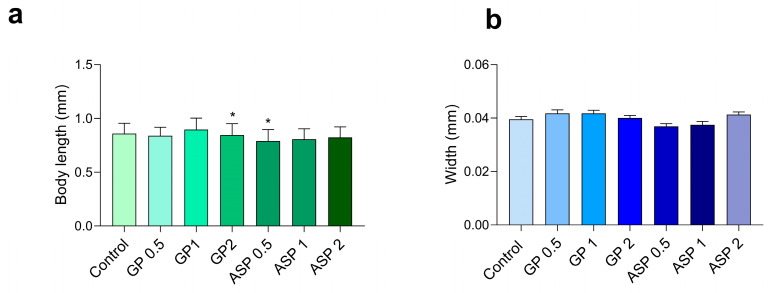
Body size (**a**) and width (**b**) of worms treated for 48 h with GP or ASP (0.5, 1 or 2 mM). N = 10 worms. * *p* < 0.05 versus control group. Values represent means ± SEM. Analyzed by one-way ANOVA followed by Dunnett’s test.

**Figure 7 biomolecules-15-01105-f007:**
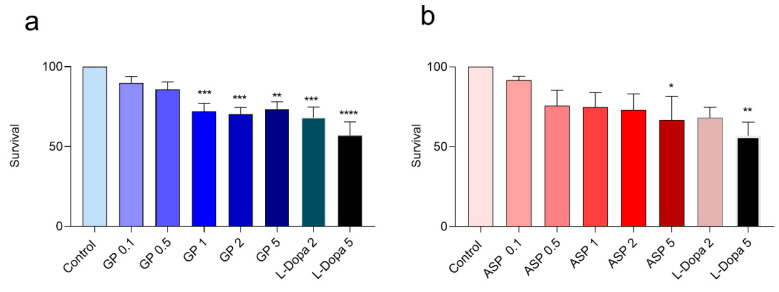
Worms’ survival rates treated with crescent concentration of GP (**a**), ASP (**b**), or L-dopa (**a**,**b**) for 24 h. N = 10 worms. Values represent means ± SEM, * *p* < 0.05, ** *p* < 0.01, *** *p* < 0.001, **** *p* < 0.0001 versus control group. Data analyzed by ANOVA followed by Dunnett’s test.

**Figure 8 biomolecules-15-01105-f008:**
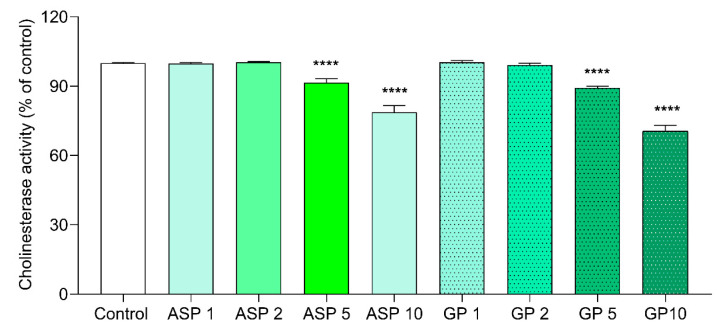
Determination of anticholinesterase activity in worms after different treatments (ASP and GP). Data are expressed as mean ± SEM (N = 28 samples with ~1000 worms). One-way ANOVA followed by Dunnett’s multiple comparisons test. Significance is indicated as follows: **** *p* ≤ 0.0001 vs. control.

**Table 1 biomolecules-15-01105-t001:** Predicted Parameter Values.

	miLogP	MW	nON	nOHNH	NV
Geniposide	−1.53	388.37	10	5	0
Asperuloside	−2.65	414.36	11	4	1
Galantamine	1.54	287.36	4	1	0
Donepezil	4.10	379.50	4	0	0
L-Dopa	−2.20	197.19	5	5	0
Apomorphine	2.89	267.33	3	2	0

miLogP—expressed as the logarithm of the octanol/water partition coefficient. MW—molecular weight. nON—hydrogen bond acceptors. nOHNH—number of hydrogen bond donors. NV—number of violations of Lipinski’s Rule of Five.

**Table 2 biomolecules-15-01105-t002:** Physicochemical parameters of geniposide and asperuloside molecules.

	TPSA	Nrotb	MV	Water Solubility (log mol/L)	Caco2 Permeability	Unbound %	VDss	Ctot	Abs (% Absorbed)
Geniposide	155.15	6	331.54	−2.81	0.35	0.66	−0.22	1.40	41.63
Asperuloside	161.22	6	340.68	−3.27	0.38	0.58	−0.19	1.21	51.52
Galantamine	41.93	1	268.19	−2.64	0.89	0.36	0.89	0.99	94.99
Donepezil	38.78	6	367.89	−4.97	1.10	0.03	1.25	1.01	93.76
L-Dopa	103.78	3	172.00	−2.89	−0.28	0.60	−0.10	0.43	47.74
Apomorphine	43.69	0	246.38	−3.94	1.18	0.16	1.18	0.98	94.22

TPSA—topological polar surface area (Å^2^). Nrotb—number of rotatable bonds. MV–molecular volume. Caco2 permeability—heterogeneous human epithelial colorectal adenocarcinoma cells. Unbound %—fraction unbound to blood proteins. VDss—volume of distribution on steady state. Ctot—total clearance. Abs—intestinal absorption (% absorbed).

**Table 3 biomolecules-15-01105-t003:** Predicted BBB-permeability parameters of GP and ASP and drugs used in AD and PD.

		AdaBust				SVM			
	Drug	MACCS	Open Babel	MolPrint	PubChem	MACCS	Open Babel	MolPrint	PubChem
	Geniposide	−0.850	1.488	−2.174	7.957	−0.031	−0.096	0.136	−0.284
	Asperuloside	0.786	−3.390	−3.198	6.057	−0.053	−0.140	0.033	−0.280
AD	Galantamine	7.509	7.849	11.736	14.128	0.154	0.091	0.649	0.346
	Donepezil	5.214	6.608	10.430	19.401	0.135	0.086	0.433	0.380
PD	L-Dopa	−2.057	−1.150	−173.760	−1.221	−0.032	−0.084	0.681	−0.112
	Apomorphine	2.326	4.772	1.996	7.093	0.071	0.063	0.288	0.222

**Table 4 biomolecules-15-01105-t004:** Binding free energy (kcal/mol) of geniposide (GP), asperuloside (ASP), and D-glucose with GLUT-1.

	GLUT-1 (kcal/mol)	Interactions
GP	−8.3	PHE26 THR30 PHE72 **ILE164** VAL165 I**LE168** GLN279 **GLN282 GLN283 ASN288PHE291 TYR292** ASN317 THR321 PHE379 GLU380 PRO383 **GLY384** PRO385 **TRP388ASN411** TRP412 **ASN415**
ASP	−7.4	PHE26 THR30 PHE72 SER73 SER80 THR137 **GLN282 GLN283ILE287 ASN288** TYR292 PHE379 **GLY384** PRO385 **TRP388** ILE404 GLY408 **ASN411** TRP412 **ASN415**
D-glucose	−5.1	ILE164 ILE168 GLN282 GLN283 ILE287 ASN288 PHE291 TYR292 PHE379 GLY384 TRP388 ASN411 ASN415

**Table 5 biomolecules-15-01105-t005:** P-gp: p-glycoprotein prediction.

	P-gp Substrate	P-gp Inhibitor
Geniposide	Yes	No
Asperuloside	Yes	No
Galantamine	No	No
Donepezil	Yes	Yes
L-dopa	Yes	No
Apomorphine	Yes	No

**Table 6 biomolecules-15-01105-t006:** Predicted Ki values of GP, ASP, and galantamine to AChE and BChE.

	AChE (Ki)	BChE (Ki)	AChE (Ki)/BChE (Ki)
GP	9.3	9.2	1.0109
ASP	9.5	9.4	1.0106
Galantamine	10	8	1.2500

**Table 7 biomolecules-15-01105-t007:** Binding free energy (kcal/mol) of GP, ASP, galantamine, and donepezil with the protein targets. AChE: acetylcholinesterase; BChE: butyrylcholinesterase.

	AChE (PDB: 4EY6)	BChE (PDB: 4TPK)
Geniposide	−8.8	−9.1
Asperuloside	−8.0	−8.7
Galantamine	−10.3	−8.6
Donepezil	−10.3	−9.5

## Data Availability

Data will be made available on request.

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
