# Peer review of "In Silico and In Vivo Pharmacological Evaluation of Iridoid Compounds: Geniposide and Asperuloside Profile Study Through Molecular Docking Assay and in the Caenorhabditis elegans Model"

_biomolecules, 2025, doi:10.3390/biom15081105_

Round 1
Reviewer 1 Report
Comments and Suggestions for Authors
This study investigates the neuroprotective potential of the iridoid compounds geniposide (GP) and asperuloside (ASP) using in silico and in vivo approach with Caenorhabditis elegans. The authors of the manuscript found that while these two compounds may have limited passive permeability across the blood-brain barrier, they might be utilized as an active transport mechanism via GLUT-1 due to their glucose moieties. Both GP and ASP demonstrated inhibitory effects on acetylcholinesterase and butyrylcholinesterase activity, and positively modulated antioxidant pathways in C. elegans, with GP affecting sod-3 expression and ASP influencing gst-4 expression in response to oxidative stress. Crucially, the compounds exhibited very high safety, both computationally and in the worm model. This research highlights GP and ASP as promising candidates for neuroprotective applications. Considering the workload, the meaning, and the creativity of the manuscript, it is recommended to be accepted by the journal Biomolecules. Here are some concerns for the authors to improve the quality of the manuscript:
- When reading the title, in silico and in vivo reads repeated corresponding to molecular docking and C. elegans model. Therefore, the title of the manuscript is recommended to be revised as: In silico and in vivo pharmacological evaluation of iridoids compounds: geniposide and asperuloside profile study: molecular docking and Caenorhabditis elegans model analysis. However, the suggestion is not in necessity if the authors insist on the original version.
- In the Abstract section, line 27, the word “reversed” should be revised as “reverse”.
- In the Keywords section, the sequence and the key words should be: neurodegeneration; iridoids; geniposide; asperuloside; molecular docking; Caenorhabditis elegans.
- In the 6th paragraph of the Introduction, the authors mentioned Chinese medicine. Reference should be added to support the viewpoint. Besides, the sentence should be revised as “Geniposide (GP) is one of the main iridoid glycosides, it has been isolated from almost 40 plant species and most of them are used as herbal medicines used in traditional Chinese medicine (TCMs).”
- In section 2.1.2, line 125, the blood-brain barrier should be revised as BBB. And the “blood-brain barrier (BBB)” should be revised as “BBB”.
- In line 155 and line 156, the origins of Marvin JS platform and Avogadro software should be provided.
- In section 2.2.3, one temperature symbol should be revised. And the origin of microscope should be provided. The version of ImageJ should be provided.
- The authors lost the title 3.1.1. And in the legend of Figure 1, the version of ChemDraw should be provided.
- The authors should discuss the limitations and prospects of the work at the end of the Discussion section. For example, the authors just mentioned the possible mechanism (GLUT-1) without any bioassay.
- For the microscope pictures from Figure 4 and Figure 5, the authors are suggested to add rulers or mention the field in the legends.
Author Response
Comment 1: When reading the title, in silico and in vivo reads repeatedly corresponding to molecular docking and C. elegans model. Therefore, the title of the manuscript is recommended to be revised as: In silico and in vivo pharmacological evaluation of iridoids compounds: geniposide and asperuloside profile study: molecular docking and Caenorhabditis elegans model analysis. However, the suggestion is not in necessity if the authors insist on the original version. In the Abstract section, line 27, the word "reversed" should be revised as "reverse".
Response 1: We appreciate your suggestion. We have implemented the corresponding change on line 27.
Comment 2: In the Keywords section, the sequence and the key words should be: neurodegeneration; iridoids; geniposide; asperuloside; molecular docking; Caenorhabditis elegans.
Response 2: Keywords have been reordered
Comment 3: In the 6th paragraph of the introduction, the authors mentioned Chinese medicine. Reference should be added to support the viewpoint. Besides, the sentence should be revised as "Geniposide (GP) is one of the main iridoid glycosides, it has been isolated from almost 40 plant species and most of them are used as herbal medicines used in traditional Chinese medicine (TCMs)."
Response 3: We appreciate your suggestion and have implemented the changes accordingly.
Comment 4: In section 2.1.2, line 125, the blood-brain barrier should be revised as BBB. And the "blood-brain barrier (BBB)" should be revised as "BBB".
Response 4: the requested changes have been made.
Comment 5: In line 155 and line 156, the origins of Marvin JS platform and Avogadro software should be provided.
Response 5: the corresponding links have been added.
Comment 6: In section 2.2.3, one temperature symbol should be revised. And the origin of microscope should be provided. The version of lmageJ should be provided.
Response 6: The temperature symbol, the origin of the microscope, and the version of the ImageJ program have been included.
Comment 7: The authors lost the title 3.1.1. And in the legend of Figure 1, the version of ChemDraw should be provided.
Response 7: The section has been renamed and the ChemDraw version has been included in Figure 1.
Comment 8: The authors should discuss the limitations and prospects of the work at the end of the Discussion section. For example, the authors just mentioned the possible mechanism (GLUT-1) without any bioassay.
Response 8: Thank you for your valuable comment. We acknowledge that a major limitation of our study is that the proposed mechanism of transport across the BBB via the glucose transporter is based on in silico analysis. While computational methods provide important preliminary insights, experimental validation would be required to confirm the predicted interaction and transport. We have now included a discussion of this limitation in the revised manuscript.
Comment 9: For the microscope pictures from Figure 4 and Figure 5, the authors are suggested to add rulers or mention the field in the legends.
Response 9: A scale has been added to the figures.
Reviewer 2 Report
Comments and Suggestions for Authors
The manuscript outlines a dual approach that integrates in silico and in vivo analyses to assess the neuroprotective potential of geniposide (GP) and asperuloside (ASP) through molecular docking and Caenorhabditis elegans models. This topic is pertinent due to the pressing demand for innovative therapeutic approaches for neurodegenerative diseases. The integration of computational and experimental models represents a significant strength, while the emphasis on iridoids introduces a novel perspective.
The manuscript necessitates substantial revisions prior to consideration for publication. Several important issues concerning experimental design, data presentation, clarity, and interpretation require attention.
- The introduction is excessively general and does not present a clearly defined hypothesis or research question. The primary objective remains ambiguous: Is it to demonstrate neuroprotection, elucidate mechanisms, or compare GP and ASP?
- GP has been extensively researched, whereas ASP remains less characterised. The justification for choosing ASP, particularly in light of the absence of previous neuroprotective studies, is insufficiently articulated.
- The docking methodology in in silico studies is insufficiently detailed. Insufficient information exists regarding the parameters, controls, and validation of docking results..
- The selection of C. elegans strains and endpoints, such as the use of GFP reporters, is suitable; however, details regarding experimental design, including the number of replicates, statistical methods, blinding, and randomisation, are lacking.
- The results section is deficient in quantitative data, lacks statistical significance values, and does not include suitable graphical representation. Figures and tables are absent and not referenced.
- The discussion exaggerates the importance of the findings, particularly concerning blood-brain barrier permeability and neuroprotective potential, in light of the limitations of the models employed.
- The manuscript proposes GLUT-1-mediated transport through docking analysis, yet lacks experimental validation.
Author Response
Comment 1: The introduction is excessively general and does not present a clearly defined hypothesis or research question. The primary objective remains ambiguous: Is it to demonstrate neuroprotection, elucidate mechanisms, or compare GP and ASP?
Response 1: Thank you for all the insightful comments. We agree that the original introduction lacked clarity regarding the primary objective and research question. In response, we have revised the introduction to explicitly state the aim of the study, which is to investigate the potential mechanisms of action by which the compounds GP and ASP may be useful in the treatment of central nervous system (CNS) disorders, as well as to evaluate their toxicity profiles. These objectives are now clearly outlined, and the revised version also reflects the rationale behind using both in silico and in vivo approaches. We hope this modification addresses your concern and improves the focus of the manuscript.
Comment 2: GP has been extensively researched, whereas ASP remains less characterised. The justification for choosing ASP, particularly in light of the absence of previous neuroprotective studies, is insufficiently articulated.
Response 2: We acknowledge that GP has been more extensively studied compared to ASP. However, the limited data available on ASP, particularly regarding its potential neuroprotective effects, was precisely the rationale for including it in our study. We believe that exploring less-characterized compounds like ASP may uncover new therapeutic possibilities and contribute to a broader understanding of their pharmacological potential. We have now clarified this point in the revised introduction to better justify the inclusion of ASP.
Comment 3: The docking methodology in in silico studies is insufficiently detailed. Insufficient information exists regarding the parameters, controls, and validation of docking results.
Response 3: We thank the reviewer for this comment. However, we respectfully note that the docking methodology is described in detail in the Methods section of the manuscript (page 4, line 150).
Comment 4: The selection of C. elegans strains and endpoints, such as the use of GFP reporters, is suitable; however, details regarding experimental design, including the number of replicates, statistical methods, blinding, and randomisation, are lacking.
Response 4: We appreciate the reviewer’s insightful comment. The information regarding the selection of C. elegans strains, the evaluated endpoints, the number of biological and technical replicates, and the statistical analyses applied is already provided in the manuscript, as detailed below:
- The number of replicates is described in the “Materials and Methods” section ‘2.2.3 Antioxidant properties’ (page 5). “Ten worms per treatment (n= 10) were analyzed....”
- The statistical methods used are detailed at the end of section 2.2.7 (page 5). “...the test solutions and the control were analyzed using one-way ANOVA, followed by Dunnett’s test…..”
- Information regarding blinding and randomization has now been added to the revised version of the manuscript. Specifically, we have clarified that all relevant analyses were conducted under blind conditions, and that group allocation was randomized to reduce potential bias. This statement has been included in the "Materials and Methods" section.
Comment 5: The results section is deficient in quantitative data, lacks statistical significance values, and does not include suitable graphical representation. Figures and tables are absent and not referenced.
Response 5: We appreciate the reviewer’s attention to the presentation of our results. We respectfully note that the current version of the manuscript already contains the requested quantitative information and visual materials. Each figure contains the sample N and the significance used, and the statistical tests are explained in the methodology section. All figures have captions and are cited sequentially according to the journal's standards. Because these elements are already present, we would be grateful if the reviewer could indicate any specific dataset or statistic they find unclear or insufficient, so we can address it promptly.
Comment 6: The discussion exaggerates the importance of the findings, particularly concerning blood-brain barrier permeability and neuroprotective potential, in light of the limitations of the models employed.
Response 6: We appreciate the reviewer’s critical perspective on the interpretation of our findings. We agree that the limitations of the models employed must be acknowledged. In response, we have revised the Discussion section to better contextualize our conclusions, clearly stating the limitations of the in vivo models used and the need for further validation in higher-order systems.
However, we respectfully emphasize that, despite these limitations, the results provide valuable preliminary evidence supporting the compound's potential biological activity. We believe that these findings contribute meaningfully to the current knowledge and justify further investigation in more complex models.
A revised paragraph has been added to the Discussion to address these points and moderate the tone of our interpretations.
Comment 7: The manuscript proposes GLUT-1-mediated transport through docking analysis, yet lacks experimental validation.
Response 7: We acknowledge the reviewer’s observation and agree that our study does not include experimental in vitro or in vivo validation of the proposed GLUT-1-mediated transport. Nonetheless, we believe that the docking analysis provides a valuable initial indication of a possible interaction with GLUT-1, which may serve as a foundation for future functional assays. This approach aligns with the exploratory nature of the study and helps guide subsequent research efforts aimed at validating the predicted interaction experimentally.
Reviewer 3 Report
Comments and Suggestions for Authors
The current study presents interesting preliminary findings on GP and ASP. Addressing the points given below would significantly improve the paper:
-The description of aducanumab's approval as "(withdrawal by the laboratory)" is somewhat informal for scientific literature.
-When discussing istradefylline's approval, stating "In certain countries" is vague.
-A brief mention of the specific type of chromatography used would be helpful for the method for extracting GP and ASP is described as "by chromatographic methods".
-There are minor grammatical errors, such as "iridoid" which should be "iridoids" , "a in silico" instead of "an in silico" , and "could reversed" which should be "could reverse" or "reversed".
-The abstract states that "GP at 1 and 2 mM could reversed the H2O2-induced increase in sod-3 expression," but the "Antioxidant properties" method section only mentions exposing worms to "GP (2 mM)". It's unclear if 1 mM was also tested.
-The instruction "see next section" for the CBDock2 platform in the "Prediction of the GP and ASP transport mechanism at BBB" section is imprecise.
-In Table 2, the column header for "Water Solubility" lacks units.
-The sample size of 10 worms per well, with experiments conducted in triplicate (totaling 30 worms per condition), might be considered a bit low for a robust statistical analysis. Can you fix this?
-Is it Cholinesterase Activity or Anticholinesterase Activity? Please correct.
-Figures 2 and 3 must be more distinctive with a better resolution. Bondings are not clear.
-In the "Docking studies" subsection (2.1.4), the sentence "GP, ASP, and galantamine the reference ligand used as an anticholinesterase control-were designed using the Marvin JS platform and converted into 3D structures in 158 Avogadro software using the Open Babel 2.3.2 tool" is repeated verbatim twice within the same paragraph.
-What was the purity of the compounds?
-"p" (statistical) should be italicized in each case.
Author Response
Comment 1: The description of aducanumab's approval as "(withdrawal by the laboratory)" is somewhat informal for scientific literature.
Response 1: We appreciate your suggestion and have implemented the changes accordingly.
Comment 2: When discussing istradefylline's approval, stating "In certain countries" is vague.
Response 2: We appreciate your suggestion and have implemented the changes accordingly.
Comment 3: A brief mention of the specific type of chromatography used would be helpful for the method for extracting GP and ASP is described as "by chromatographic methods".
Response 3: Thank you for the suggestion. We have included a brief description of the specific chromatographic methods used for the extraction of GP and ASP in the revised manuscript.
Comment 4: There are minor grammatical errors, such as "iridoid" which should be "iridoids", "a in silico" instead of "an in silico", and "could reversed" which should be "could reverse" or "reversed".
Response 4: We appreciate your suggestion and have implemented the changes accordingly.
Comment 5: The abstract states that "GP at 1 and 2 mM could reversed the H2O2-induced increase in sod-3 expression," but the "Antioxidant properties" method section only mentions exposing worms to "GP (2 mM)". It's unclear if 1 mM was also tested.
Response 5: We appreciate your suggestion and have implemented the changes accordingly.
Comment 6: The instruction "see next section" for the CBDock2 platform in the "Prediction of the GP and ASP transport mechanism at BBB" section is imprecise.
Response 6: We appreciate your suggestion and have implemented the changes accordingly.
Comment 7: In Table 2, the column header for "Water Solubility" lacks units.
Response 7: We appreciate your suggestion and have implemented the changes accordingly.
Comment 8: The sample size of 10 worms per well, with experiments conducted in triplicate (totaling 30 worms per condition), might be considered a bit low for a robust statistical analysis. Can you fix this?
Response 8: We appreciate the reviewer’s observation regarding sample size. However, at this stage, it is not feasible to increase the number of animals due to logistical and ethical constraints. It is important to highlight that our experimental design, using 10 worms per well in triplicate (30 worms per condition), is consistent with established protocols reported in several peer-reviewed studies employing C. elegans for similar purposes (e.g., [Azevedo et al., 2019(https://doi.org/10.3390/molecules24183299)]; [Hamdi et al., 2025]( https://doi.org/10.3390/foods14061048)). Furthermore, this design has been sufficient to detect statistically significant differences in previous experiments using similar endpoints and methodologies.
Comment 9: Is it Cholinesterase Activity or Anticholinesterase Activity? Please correct.
Response 9: We appreciate your suggestion and have implemented the changes accordingly.
Comment 10: Figures 2 and 3 must be more distinctive with a better resolution. Bondings are not clear.
Response 10: We appreciate your suggestion and have implemented the changes accordingly.
Comment 11: In the "Docking studies" subsection (2.1.4), the sentence "GP, ASP, and galantamine the reference ligand used as an anticholinesterase control-were designed using the Marvin JS platform and converted into 3D structures in 158 Avogadro software using the Open Babel 2.3.2 tool" is repeated verbatim twice within the same paragraph.
Response 11: We appreciate your suggestion and have implemented the changes accordingly.
Comment 12: What was the purity of the compounds?
Response 12: We thank the reviewer for the observation. Unfortunately, we do not have specific analytical data on the purity of the isolated compounds, but it is expected to be higher than 95%.
Comment 13: "p" (statistical) should be italicized in each case.
Response 13: We appreciate your suggestion and have implemented the changes accordingly.
Round 2
Reviewer 1 Report
Comments and Suggestions for Authors
The auhtors have replied the comments, and the present version of the manuscript reads better than the last verision. The present version of the manuscript is recommended to be accepted by the journal Biomolecules. Congratulations to the authors.
Reviewer 3 Report
Comments and Suggestions for Authors
The authors have greatly improved the article. Although their answers to some of the criticisms are not adequate, it seems appropriate to accept the article as it stands.